# Quadratic Quantum Variational Monte Carlo

**Baiyu Su**
University of Texas at Austin
`baiyusu@utexas.edu`

**Qiang Liu**
University of Texas at Austin
`lqiang@cs.utexas.edu`

## Abstract

This paper introduces the Quadratic Quantum Variational Monte Carlo ($Q^2$VMC) algorithm, an innovative algorithm in quantum chemistry that significantly enhances the efficiency and accuracy of solving the Schrödinger equation. Inspired by the discretization of imaginary-time Schrödinger evolution, $Q^2$VMC employs a novel quadratic update mechanism that integrates seamlessly with neural network-based ansatzes. Our extensive experiments showcase $Q^2$VMC's superior performance, achieving faster convergence and lower ground state energies in wavefunction optimization across various molecular systems, without additional computational cost. This study not only advances the field of computational quantum chemistry but also highlights the important role of discretized evolution in variational quantum algorithms, offering a scalable and robust framework for future quantum research.

## 1 Introduction

Finding fast and accurate approaches to solving Schrödinger equations is a central challenge in quantum chemistry, with far-reaching implications for material science and pharmaceutical development. The ability to solve this equation precisely would unlock a plethora of properties inherent to the microscopic systems being studied. However, the task of deriving exact wavefunctions for even moderately sized molecules is notoriously difficult, with no analytical solutions in general cases.

The advent of deep learning has significantly advanced the field of quantum chemistry, particularly through enhancements in the *Quantum Variational Monte Carlo (QVMC)* method [1–3]. Enhanced by neural network-based approaches, commonly referred to as *neural ansatz*, methods like PauliNet [4] and FermiNet [5, 6] have demonstrated remarkable success. These approaches often match or surpass the accuracy of traditional "gold standard" methods such as CCSD(T) [7] even for complex molecules [8, 9]. This rapid development has spurred a broad spectrum of research into more accurate and efficient neural ansatz models, significantly impacting ab-initio quantum chemistry [10–12]. Recent reviews [13] provide comprehensive overviews of the advancements and diverse applications extending beyond molecular systems to areas like solid-state physics and electron gases [14–16].

Despite the accuracy and flexibility of Quantum Variational Monte Carlo (QVMC), optimizing it remains a challenging task, often requiring prolonged convergence times. Various methods have been developed to accelerate training, such as stochastic reconfiguration (SR) [17–19], Newton method [20], adaptive imaginary-time evolution [21], and Wasserstein Quantum Monte Carlo (WQMC) [22]. In our work, we enhance optimization efficiency by employing the perspective of imaginary-time Schrödinger evolution [23, 24], which naturally guides the wavefunction toward the ground state over an extended time horizon. According to McLachlan's variational principle, it can be shown that this continuous-time process yields parametric updates analogous to those in standard QVMC with infinitesimal learning rates [25]. However, while theoretically robust, implementing this evolution in practical settings is challenging with finite time steps. Traditional approaches approximate the updates within parametric space, but this method is limited by the non-convex nature of the objective and the unpredictability arising from complex theoretical properties. To overcome these challenges, we propose discretizing the evolution process itself, ensuring convergence to the ground state even

---

**Algorithm 1** QVMC vs Q$^2$VMC

---

**Require:** Molecule Hamiltonian $\hat{H}$, a neural ansatz $\psi_\theta(\mathbf{x})$ of wavefunction parameterized by $\theta$
**Require:** Initial weights $\theta_0$, an optimizer `optimizer`, and learning rate schedule $\{\eta_t\}_{t=0}^{T-1}$
  **while** not converged **do**
    Draw sample $\{\mathbf{x}^{(i)}\colon i = 1, \ldots, N\}$ from $\psi_{\theta_t}^2(\mathbf{x})$ via MCMC. Calculate local energy and loss:

$$E_L(\mathbf{x}^{(i)}) = \psi_{\theta_t}^{-1}(\mathbf{x}^{(i)})\hat{H}\psi_{\theta_t}(\mathbf{x}^{(i)}), \qquad \mathcal{L}(\theta_t) = \frac{1}{N}\sum_{i=1}^{N} E_L(\mathbf{x}^{(i)}),$$

Update model weights $\theta$ via $\theta_{t+1} = \texttt{optimizer}(\theta_t, \Delta\theta, \tilde{F})$ (see Eq. 19), where

$$\Delta\theta = -\frac{1}{N}\sum_i \left( c^{(i)} - \frac{1}{N}\sum_j c^{(j)} \right) \nabla_\theta \log \psi_{\theta_t}(\mathbf{x}^{(i)})$$

QVMC:    $c^{(i)} = \eta_t E_L(\mathbf{x}^{(i)})$,    Q$^2$VMC:    $c^{(i)} = \eta_t E_L(\mathbf{x}^{(i)}) - \frac{1}{2}\eta_t^2 E_L^2(\mathbf{x}^{(i)})$

  **end while**
  **return** the neural wavefunction $\psi_{\theta_T}^2(x)$, and samples $\{x^{(i)}\}_{i=1}^{N} \sim \psi_{\theta_T}^2(x)$

---

with finite time steps. We then project the discretely evolved distribution back into parametric space, forming an update algorithm that iteratively refines the neural ansatz towards the ground state.

Diffusion Monte Carlo (DMC) [26–28] is a well known method in quantum chemistry that also employs ground state projection. Known for its promising results, DMC often surpasses the limitations of specific ansatz choices [29–32]. However, as a non-parametric approach, DMC offers flexibility and computational efficiency but lacks the ability to provide explicit values of the wavefunction, which can be essential in applications. Additionally, DMC methods encounter the fixed-node approximation issue: their effectiveness depends heavily on the accuracy of a fixed trial wavefunction, which cannot be improved during the computation. By contrast, our approach maintains a parametric representation of the wavefunction that evolves continuously toward the ground state, effectively sidestepping the limitations posed by fixed-node constraints.

A few previous works have similarly focused on projecting an evolved quantum state onto the parametric manifold of an ansatz, as explored in [22, 33]. To the best of our knowledge, all existing approaches rely on conventional projection methods, specifically the quantum fidelity or the Fubini-Study metric. Although these metrics are widely used in physics, their mathematical properties are intricate and remain underexplored [34]. Furthermore, none of these methods account for finite step size. In contrast, our approach takes advantage of the fact that wavefunction analysis is primarily conducted through the probability distribution ($q \propto |\psi|^2$) derived from it. Accordingly, we project probability distributions using the Kullback-Leibler divergence, chosen for its mathematical simplicity and its ability to effectively capture distributional differences of interest. The introduction of a quadratic term naturally emerges from the squared nature of the wavefunction in the probability distribution, while the preconditioning by the Fisher information matrix arises naturally from the curvature of this projection.

Building on this framework, we introduce the *Quadratic Quantum Variational Monte Carlo (Q$^2$VMC)*, an innovative optimization mechanism that enhances the conventional QVMC by allowing finite-time updates without additional computational overhead, as detailed in Algorithm 1. This novel approach not only maintains theoretical equivalence with QVMC under infinitesimally small time steps but also demonstrably achieves **twice the optimization speed / significantly better accuracy** within the same computational budget.

## 2 Results

In this section, we present a brief overview of the results achieved by the Quadratic Quantum Variational Monte Carlo ($Q^2$VMC) method, demonstrating its enhanced efficiency and accuracy in wavefunction optimization. Our method achieves improvements in convergence speed and energy accuracy across various molecular systems. Technical details about the relevant experiments is written in the experiments section.

**Summary of key results** We evaluated the performance of $Q^2$VMC against traditional Quantum Variational Monte Carlo (QVMC) using state-of-the-art attention based neural network ansatzes: Psiformer [8] and LapNet [9]. A total of six different molecules with diverse sizes are tested, with number of electrons ranging from 6 to 30 to demonstrate robustness. Each one of the 12 possible combinations are optimized with the default settings as in their original papers where possible and with (our method) or without (baseline reproduce) the quadratic modification. Our findings indicate that $Q^2$VMC not only accelerates the convergence process but also reduces the variance in batch energies, suggesting a more stable approach towards reaching the ground state. These enhancements are highlighted as:

- **Faster Convergence:** As demonstrate by the training curves in Figure 1 $Q^2$VMC shows a 2x speed-up in optimization comparing with the baselines, achieving the target energies in approximately half the iterations required by QVMC.

- **Enhanced Accuracy:** The energy accuracies obtained are consistently superior to those achieved by conventional QVMC, as detailed in Table of energy accuracies 8. This superiority is particularly pronounced in complex systems with a higher number of electrons, where the traditional methods struggle to maintain precision and stability.

- **Simple Integration and Hyperparameter Robustness:** As shown in Algorithm 1, $Q^2$VMC can be seamlessly incorporated into existing frameworks with only a single line of code change. This section presents results obtained with the original hyperparameters, highlighting that effective performance gains are achievable without additional tuning efforts. For completeness, Appendix C provides results from experiments where hyperparameters were adjusted specifically for $Q^2$VMC, showing that these tuned settings achieve comparable or superior performance to the Psiformer (Large) model using the traditional QVMC method, despite the latter's use of a network approximately four times larger than the Psiformer (Small) employed here.

Table 1: Energies for a set of molecules studied in Psiformer [8] and LapNet [9]. Reference energies are taken from the respective papers. In order to eliminate any potential effects from different evaluation strategies, we also report our reproduced baseline values in the appendix.

| System (Electrons) | Psiformer | $Q^2$VMC+Psi | LapNet | $Q^2$VMC+Lap |
|---|---|---|---|---|
| $Li_2$ (6) | -14.99486(1) | -14.99490(1) | -14.99485(1) | -14.99486(1) |
| $NH_3$ (10) | -56.56367(2) | -56.56374(2) | -56.56359(2) | -56.56370(2) |
| CO (14) | -113.32416(4) | -113.32442(2) | -113.32417(4) | -113.32428(2) |
| $CH_3NH_2$ (18) | -95.86050(4) | -95.86073(2) | -95.86025(3) | -95.86053(2) |
| $C_2H_6O$ (26) | -155.04656(7) | -155.04696(3) | -155.04563(6) | -155.04619(4) |
| $C_4H_6$ (30) | -155.94619(8) | -155.94665(4) | -155.94528(4) | -155.94618(4) |

## 3 Background

### 3.1 Quantum Variational Monte Carlo (QVMC)

At the heart of quantum mechanics lies the *wavefunction*, which embodies all possible classical states of a system. When first quantization is considered, the wavefunction serves as a mapping from the states of particles to complex amplitudes. For instance, the state of a single electron can be represented by its position $\mathbf{x} \in \mathbb{R}^3$ and spin $\sigma \in \{\uparrow, \downarrow\}$. Consequently, the wavefunction of an $N$-electron system is a mapping $\psi : \left(\mathbb{R}^3 \times \{\uparrow, \downarrow\}\right)^N \to \mathbb{C}$, with the square of its magnitude, $|\psi|^2$,

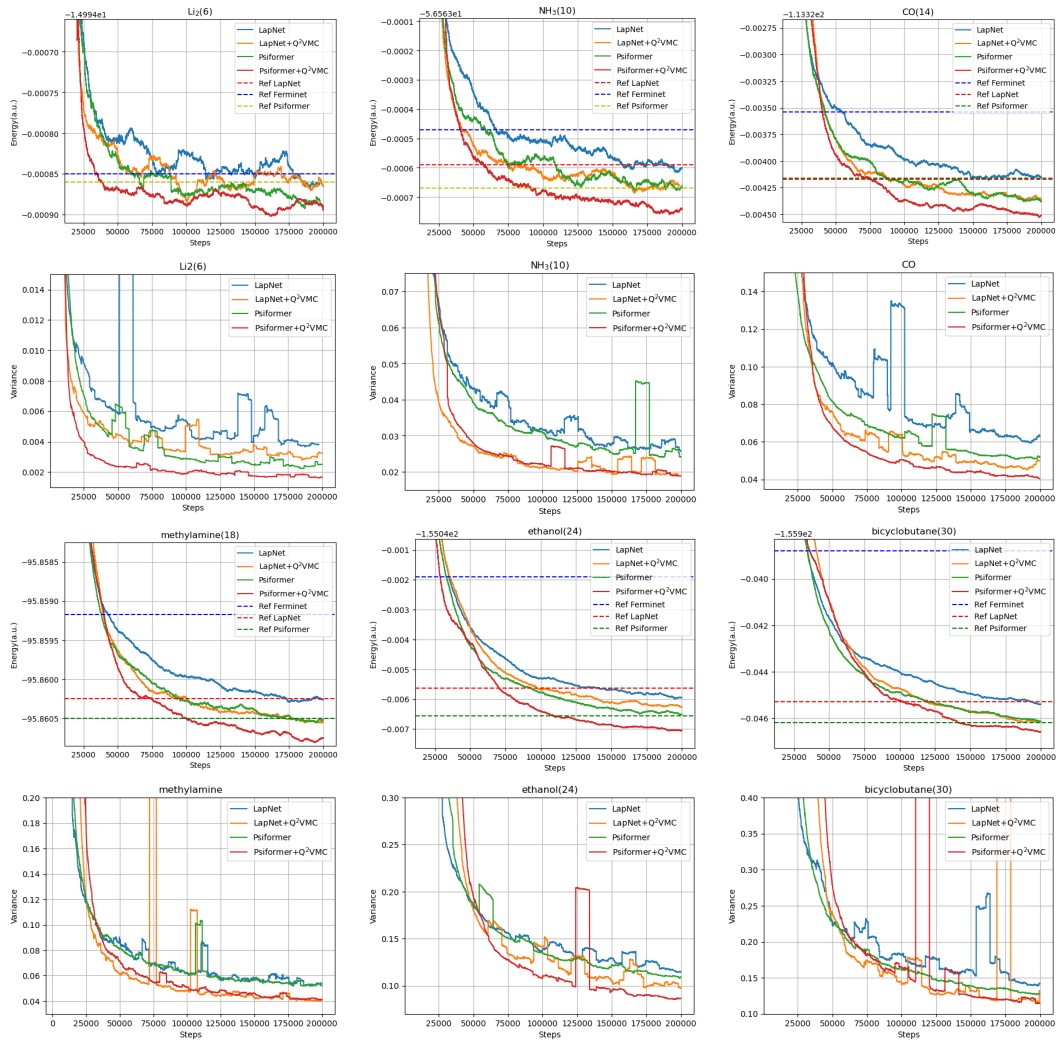

Figure 1: Optimization curves for different molecules

representing a probability density $\pi_{\psi^2} = |\psi|^2 / C$, where $C = \int |\psi|^2$ is the normalization constant. The probability density $\pi_{\psi^2}$ represents the likelihood of observing the quantum system in a specific state upon measurement. Note that when the normalization condition is not enforced, wavefunctions $\psi$ are invariant under scalar multiplication, implying $\psi \sim a\psi$ for any non-zero scalar $a$, where all such functions correspond to the identical normalized probability density $\pi_\psi$. With an abuse of notation, we simply write $\pi_{\psi^2} = |\psi|^2$ when corresponding normalization is clear from the context.

The behavior of non-relativistic quantum systems are dictated by the Schrödinger equation, which, in its time-independent form, poses an eigenfunction problem $\hat{H}\psi = E\psi$. Here, $\hat{H}$ represents the Hermitian linear operator known as the Hamiltonian, and the eigenvalue $E$ represents the energy associated with a specific eigenfunction. The Hamiltonian's structure is crucial, encapsulating the physical properties of the quantum system. In quantum chemistry, the Hamiltonian generally takes the form of:

$$\hat{H} = -\frac{1}{2}\nabla_{\mathbf{x}}^2 + V(\mathbf{x}), \tag{1}$$

where $\nabla_{\mathbf{x}}^2$ is the Laplacian in coordinate space, and $V(\mathbf{x})$ represents a potential function dependent on the particle positions (e.g. configurations of the nucleus).

Quantum Variational Monte Carlo (QVMC) is a computational approach used to determine the ground state, corresponding to the lowest eigenvalue $E_0$, of the Schrödinger equation. In the studies

of QVMC, two simplifications are commonly employed. First, given that $\hat{H}$ is Hermitian, its eigenfunctions $\psi$ can be considered real-valued, permitting a focus solely on real-valued wavefunctions. Second, the absence of spin variables $\sigma_i$ in the Hamiltonian allows for simplification in modeling the system, permitting us to fix the spins of electrons and shift our focus solely on their positional values. Hence, in QVMC, we utilize an *unnormalized* wavefunction ansatz $\psi_\theta : \mathbb{R}^{3N} \to \mathbb{R}$, parameterized by $\theta$. The term *neural ansatz* denotes the representation of $\psi_\theta$ by a neural network.

The quest for the ground state solution is guided by the Rayleigh-Ritz principle, which involves minimizing the loss function:

$$\mathcal{L}(\theta) = \frac{\int \psi_\theta(\mathbf{x})\hat{H}\psi_\theta(\mathbf{x})\mathrm{d}\mathbf{x}}{\int |\psi_\theta(\mathbf{x})|^2 \, \mathrm{d}\mathbf{x}} = \mathbb{E}_{|\psi_\theta|^2}[\underbrace{\psi_\theta^{-1}(\mathbf{x})\hat{H}\psi_\theta(\mathbf{x})}_{=E_{L,\theta}(\mathbf{x})}] \geq E_0, \tag{2}$$

where $E_{L,\theta}(\mathbf{x}) \overset{def}{=} \psi_\theta^{-1}(\mathbf{x})\hat{H}\psi_\theta(\mathbf{x})$ represents the *local energy*. The gradient of this loss function is computed as

$$\nabla_\theta \mathcal{L}(\theta) = \mathbb{E}_{|\psi_\theta|^2} \left[ \left( E_L(\mathbf{x}) - \overline{E_L} \right) \nabla_\theta \log \psi_\theta^2(\mathbf{x}) \right], \tag{3}$$

with $\overline{E_L} = \mathbb{E}_{|\psi^2|}[E_L(\mathbf{x})]$ denoting the average local energy. Minimizing $\mathcal{L}$ with gradient descent thus yields an iterative process, where Markov Chain Monte Carlo sampling is employed to extract samples from distribution $|\psi_\theta|^2$ and the samples are used estimate the energy gradient, which then directs the parameter updates [35].

To improve the efficiency of optimization, the *Stochastic Reconfiguration* [36, 17] or Quantum Natural Gradient Descent [37, 38]—has commonly been adopted for QVMC updates. This method enhances convergence by preconditioning the gradient with an (approximation of) Fisher information matrix $\hat{F}(\theta)^{-1}$ related to the quantum state $\psi_\theta^2$, which can be implemented efficiently using approximate natural gradient frameworks like KFAC [39]. Consequently, the practical parameter update step, considering a learning rate $\eta$, is given as

$$\Delta\theta_{\mathrm{QVMC}} = \eta\hat{F}(\theta)^{-1}\nabla_\theta\mathcal{L}(\theta). \tag{4}$$

## 4 Quadratic Quantum Variational Monte Carlo

This section introduces our methodology for updating the neural ansatz through the Q$^2$VMC approach. In Section 4.1, we present a discretized imaginary-time Schrödinger evolution. This process operates within the non-parametric Hilbert space of wavefunctions, guiding the system progressively toward the ground state. Subsequently, in section 4.2, we discuss how to project the evolved distributions back onto the parametric manifold of the neural ansatz by minimizing the Kullback-Leibler divergence between the evolved and updated distributions, which forms the basis of the Q$^2$VMC algorithm.

### 4.1 Imaginary-Time Schrödinger Evolution

Consider a Hilbert space equipped with the inner product $\langle u, v \rangle = \int uv$, and spanned by orthonormal basis functions $\{\phi_i(\mathbf{x})\}$. Given a Hermitian operator $\hat{H}$, normalizing its eigenfunctions so that $\langle \phi_i, \phi_i \rangle = 1$, results in a basis that embodies three essential attributes: 1) they are eigenfunctions of the Hamiltonian with associated eigen-energies $E_i$, 2) normalized, and 3) mutually orthogonal for distinct indices. These attributes ensure that any function within our Hilbert space can be precisely represented as linear combinations of these orthonormal basis functions associated with the Hamiltonian. The energies of these eigenfunctions are conventionally ordered as $E_0 < E_1 < E_2 < \cdots$. Thus, the primary objective of QVMC is to approximate $\phi_0$, with is associated to the lowest energy $E_0$, using a parametric ansatz $\psi_\theta$.

The imaginary-time Schrödinger equation emerges from the time-dependent Schrödinger equation by substituting $t'$ with $-it$, yielding:

$$-\frac{\partial\psi(\mathbf{x}, t)}{\partial t} = \hat{H}\psi(\mathbf{x}, t). \tag{5}$$

Considering an initial wavefunction at $t = 0$ as $\psi(\mathbf{x}, t = 0) = \sum_{i=0}^{\infty} \alpha_i \phi_i$, the imaginary-time Schrödinger evolution has a closed-form solution expressed in terms of these basis functions:

$$\psi(\mathbf{x}, t) = e^{-t\hat{H}} \sum_{i=0}^{\infty} \alpha_i \phi_i(\mathbf{x}) = \sum_{i=0}^{\infty} \alpha_i e^{-tE_i} \phi_i(\mathbf{x}).$$

Scaling the wavefunction by a factor of $e^{tE_0}$ reveals the evolution's impact:

$$\tilde{\psi}(\mathbf{x}, t) = \phi_0(\mathbf{x}) + \sum_{i=1}^{\infty} \frac{\alpha_i}{\alpha_0} e^{-t(E_i - E_0)} \phi_i(\mathbf{x}). \tag{6}$$

Given $E_i - E_0 > 0$ for all $i > 0$, as $t \to \infty$, the wavefunction's projection onto any basis other than $\phi_0$ diminishes to zero. Hence, starting with any wavefunction that overlaps with $\phi_0$, the imaginary-time Schrödinger evolution consistently approximates the ground state as $t$ approaches infinity.

While in theory, the operator $e^{-t\hat{H}}$ as $t \to \infty$ can directly yield the ground state function, exact computation of this operator is impractical. One must discretize the time step to incrementally evolve the process. The evolution process can conveniently operate with discrete time in the following manner.

**Definition 4.1** (Discretization). For a given time step $\tau$, the Discretized Imaginary-Time Schrödinger Evolution, corresponding to a Hamiltonian $\hat{H}$ and its ground state energy $E_0$, is described by a series of functions $\{\psi^{(n)}\}_{n=0}^{\infty}$ such that:

$$\psi^{(n+1)} = \frac{1 - \tau\hat{H}}{1 - \tau E_0} \psi^{(n)} = \frac{1 - \tau E_L^{(n)}}{1 - \tau E_0} \psi^{(n)}, \tag{7}$$

where $E_L^{(n)} = \hat{H}\psi^{(n)}/\psi^{(n)}$.

This process is proven to converge to the ground state:

**Theorem 4.2** (Convergence). *Assuming $\langle \psi^{(0)}, \phi_0 \rangle \neq 0$ and $\|\psi^{(0)}\|_2 < \infty$, then $\psi^{(n)}$ weakly converges to $\phi_0$, up to a constant factor, as $n \to \infty$.*

*Remark* 4.3. This result differs from its continuous time counterpart as shown in equation 6. The evolution process in our approach does not require an infinitesimal time step to converge, thus it is *insensitive* to the size of the time step taken. Asymptotic convergence is guaranteed regardless of the time step size. Consequently, unlike previous methods, our approach does not necessitate the use of very small time steps, which can often impede effective convergence.

The evolution in the Hilbert space of wavefunctions may also be motivated from other perspectives, e.g gradient flow under Fisher-Rao metric [22] and the discrete evolution is similar to the *quantum power method* in the computational quantum literature in the limit that $\tau \to \infty$ (see, for example, [40, 41]). For a more comprehensive theoretical analysis, we direct readers to these sources.

## 4.2 Parametric Projection of the Evolution Process

In practical applications, operating within the infinite-dimensional space of functions is not feasible. Instead, we utilize a neural ansatz $\psi_\theta$ parameterized by a finite set of parameters $\theta$ to approximate the underlying functional. This necessitates an iterative process: a) evolving the current $\psi_\theta$ following discrete evolution to produce $(1 - \tau E_L)\psi_\theta$, b) projecting the evolved function back into the parametric space to update model parameters, resulting in $\psi_{\theta + \Delta\theta}$, and c) updating associated MCMC data samples based on this projection before repeating the process with the updated neural ansatz.

While step a) is straightforward and efficiently implementable (as detailed in Appendix A), step b) requires a suitable divergence metric for effective projection. In this study, we minimize the Kullback-Leibler (KL) divergence between the probability distribution induced by the evolved wavefunction and that represented by the updated neural ansatz within a trust region [42]:

**Proposition 4.4.** *Let $h(\Delta\theta)$ denote the KL-divergence between the evolved distribution $(1 - \tau E_L(\boldsymbol{x}))^2 \psi_\theta^2(\boldsymbol{x})$ and the updated distribution $\psi_{\theta + \Delta\theta}^2$:*

$$h(\Delta\theta) = \mathcal{KL}\left[(1 - \tau E_L(\boldsymbol{x}))^2 \psi_\theta^2(\boldsymbol{x}) \| \psi_{\theta + \Delta\theta}^2(\boldsymbol{x})\right]. \tag{8}$$

*Given the size of trust region $\epsilon$, our objective of projection is*

$$\Delta\theta_\epsilon^* = \arg\min_{\Delta\theta} \left\{ h(\Delta\theta) \quad s.t. \quad \mathcal{KL}(\psi_{\theta+\Delta\theta}^2 \| \psi_\theta^2) \leq \epsilon^2/2 \right\}. \tag{9}$$

*As $\epsilon \to 0^+$, the optimal update direction approaches to*

$$\Delta\theta^* = \lim_{\epsilon \to 0^+} \frac{1}{\epsilon} \Delta\theta_\epsilon^* = -\frac{F^{-1}g}{g^\top F^{-1}g}, \tag{10}$$

*where $g$ and $F$ are the gradient and Fisher information matrix respectively:*

$$g = \mathbb{E}[\nabla_\theta \log \psi_\theta^2(\boldsymbol{x})] - \left(\mathbb{E}\left[(1 - \tau E_L(\boldsymbol{x}))^2\right]\right)^{-1} \mathbb{E}\left[(1 - \tau E_L(\boldsymbol{x}))^2 \nabla_\theta \log \psi_\theta^2(\boldsymbol{x})\right],$$

$$F = \mathbb{E}\left[\left(\nabla_\theta \log \psi_\theta^2 - \mathbb{E}\left[\nabla_\theta \log \psi_\theta^2\right]\right)\left(\nabla_\theta \log \psi_\theta^2 - \mathbb{E}\left[\nabla_\theta \log \psi_\theta^2\right]\right)^\top\right].$$

*represents the Fisher information matrix associated with the distribution induced by the neural ansatz. All expectations here are taken with respect to the distribution $|\psi_\theta(\boldsymbol{x})|^2$.*

*Proof.* Refer to Appendix B. □

Therefore, the update given by the projecting the evolution process, i.e. the $Q^2VMC$ update, is like:

$$\Delta\theta_{Q^2VMC} \propto F^{-1}\tilde{g}$$
$$\approx \hat{F}(\theta)^{-1}\left(\mathbb{E}\left[(1 - \tau E_L(\mathbf{x}))^2 \nabla_\theta \log \psi_\theta^2(\mathbf{x})\right] - \mathbb{E}\left[(1 - \tau E_L(\mathbf{x}))^2\right] \mathbb{E}\left[\nabla_\theta \log \psi_\theta^2(\mathbf{x})\right]\right)$$

Because the mean of the scaling factors $\mathbb{E}\left[(1 - \tau E_L(\mathbf{x}))^2\right]$ is subtracted from the update, the constant of 1 does not matter. Therefore, the form of update derived solely from the perspective of discretizing imaginary-time Schrödinger evolution and projection match closely with the update of QVMC upon choosing the time step $\tau = \frac{1}{2}\eta$ except for the quadratic term of $\frac{1}{2}\eta^2 E_L^2(\mathbf{x})$. Because the term $E_L(\mathbf{x})$ has already been computed in the QVMC and the quadratic term can be added on with no effort, our method has no relative computational overhead. Notably by taking the infinitesimal time step limit $\tau \to 0$ will exactly recover the QVMC update as the additional term decays with $\mathcal{O}(\tau^2)$, thereby showing the consistency of our method in the small step size regime.

## 5  Experiments

This section details the experimental setup and evaluation strategy utilized to obtains the results shown above.

**Methodology Overview**  Two recent cutting-edge, attention-based neural ansatzes, Psiformer [8] and LapNet [9], are tested in our evaluations. The architectural hyperparameters are delineated in Table 4 in Appendix. Note that [8] provides two possible model sizes, Psiformer Small and Psiformer Large with the latter roughly 4x size than the former, and our experiments are all conducted with the former. To demonstrate the easy integration and robustness of our method, we adhered to all the original training hyperparameters from their publications (detailed in Appendix Table 5). Training curves of baseline and horizontal lines representing reference energies from respective papers are plotted to facilitate comparison and indicate successful reproduction of the claimed performance. The only modification in our $Q^2VMC$ experiments pertains to the gradient coefficients, in accordance with the $Q^2VMC$ update rule 1.

**Convergence and Stability**  Figure 2 presents the energy convergence trajectories for six molecules, demonstrating that $Q^2VMC$ achieves both rapid and consistent convergence across a range of systems. Specifically, we tested on $Li_2$ (6), $NH_3$ (10), CO (14), methylamine-$CH_3NH_2$ (18), ethanol-$C_2H_6O$ (26), and bicyclobutane-$C_4H_6$ (30), where the numbers in parentheses denote electron counts.

**Training and Evaluation** : Consistent with the methodologies as in the referenced studies, we optimize the models to 200,000 training iterations for all molecular systems. Nevertheless, it was observed that smaller molecules typically can reach convergence in fewer iterations e.g. $Li_2$ while larger systems like bicyclobutane has clearly not converged yet within the duration. We encourage future benchmarking in this field to select the total iterations adaptively. We adopt similar numerical hacks as in [8, 9] to facilitate numerical stability. Importantly, the local energies are clipped so that values will be within the range $\rho = 5.0$ of mean absolute deviation from its median. The coefficients $c^{(i)}$ are then further computed after clipping.

Following the training, an additional evaluation was conducted over 20,000 steps, employing MCMC to sample batches of data without updating the network parameters. The computed energies for the tested molecules, comparing against benchmark values, are tabulated in Table 1. For smaller molecules like $Li_2$, the energy performance gains were marginal, highlighting the system-dependent aspect of convergence energy. However, our approach facilitated consistently faster convergence across the board. For larger molecules, which do not reach convergence within the allocated iterations, $Q^2VMC$ markedly improved energy performance.

## 5.1 Ablation Study

The update of $Q^2VMC$ can be decomposed into two parts:

$$\Delta\theta_{Q^2VMC} = \Delta\theta_{QVMC} + \frac{1}{2}\eta^2\mathbb{E}\left[\left(E_L^2(\mathbf{x}) - \overline{E_L^2}\right)\nabla\log\psi_\theta^2(\mathbf{x})\right] \tag{11}$$

Therefore, one might suspect that if the better performance of $Q^2VMC$ comes solely from its larger update magnitude $\|\Delta\theta_{Q^2VMC}\| \overset{?}{>} \|\Delta\theta_{QVMC}\|$ and we can make better performance of QVMC by utilizing a larger learning rate. Note that the greater relation cannot be confirmed as the terms being added are not in the same direction. In this section, we confirm that the performance of QVMC can indeed be boosted by carefully tuning for a larger learning rate within a specific system. However, one cannot make QVMC perform better than $Q^2VMC$ solely by tuning the learning rate.

We primarily study the system of $NH_3(10)$ as it is large enough to allow different algorithms distinguish while not so large to allow objectives to converge within the 200k steps duration. All experiments in this section are done with the Psiformer model. We take a fine-grained tuning of the learning rate $\eta_0$ of QVMC within the range of $\{0.01, 0.02, 0.05, 0.1, 0.2, 0.5\}$. The strategy of training and evaluation follows the experiments. The convergence energies are listed in Table 2 and the training curves can be found in Appendix C.

Table 2: Convergence energies of $NH_3$ trained with QVMC using different $\eta_0$.

| $\eta_0$ | 0.01 | 0.02 | 0.05 |
|---|---|---|---|
| $E_{\text{converge}}$ | -56.56327(3) | -56.56350(2) | -56.56366(2) |
| $\eta_0$ | 0.1 | 0.2 | 0.5 |
| $E_{\text{converge}}$ | -56.56372 (1) | -56.56379(1) | Diverge |

It is clear from the results that with larger learning rates for QVMC, one can yield better convergence energy values, while setting it to over-large values will make the training diverge, even if the gradient clipping in terms of the Fisher norms are enabled [43]. Among the trainings of different learning rates, the best performance is obtained from the one using learning rate of $\eta_0 = 0.2$. We therefore use the matched learning rate for training with $Q^2VMC$ to see if it can still do better than this. The convergence energy values are listed in Table 3.

Table 3: Convergence energies of $NH_3$ trained with $Q^2VMC$ using different $\eta_0$.

| $\eta_0$ | 0.05 | 0.2 |
|---|---|---|
| $E_{\text{converge}}$ | -56.56374(2) | -56.56384(1) |

Our method is already performing well enough even with the default learning rate of 0.05 not tuned specifically for the system of $NH_3$. Furthermore, since experiments have already shown that the performance of $Q^2VMC$ is superior to any trials obtained from optimizing the objective using QVMC, there is no need to further tune the learning rate of $Q^2VMC$ for comparison.

Following the heuristics recommended by [10], we further ablated additional hyperparameters, including increased decay time, reduced learning rate, and reduced norm constraints. The results, summarized in Table 4, specify the modified hyperparameters alongside the achieved ground state energies upon convergence. As shown, none of these adjustments matched or exceeded the accuracy attained by $Q^2VMC$.

Table 4: More experiments for ablation study: Computed ground state energies of $NH_3$ molecules with standard quantum Monte Carlo method. Tested with different (reduced) learning rates, (increased) learning rate decay times, and (increased) norm constraints.

| Learning Rate | Decay Time | Norm Constraint | Energy |
|---|---|---|---|
| 2e-2 | 1e4 | 3e-3 | -56.56349(3) |
| | | 1e-2 | -56.56366(2) |
| | | 3e-2 | Diverge |
| | 3e4 | 3e-3 | -56.56369(1) |
| | | 1e-2 | -56.56366(1) |
| | | 3e-2 | Diverge |
| 5e-2 | 1e4 | 3e-3 | -56.56371(2) |
| | | 1e-2 | -56.56313(3) |
| | | 3e-2 | Diverge |
| | 3e4 | 3e-3 | -56.56374(1) |
| | | 1e-2 | -56.56344(2) |
| | | 3e-2 | Diverge |

### 5.2 Code and Computational Details

All models were implemented using the JAX framework [44], which is available under the Apache-2.0 License. The architectures were adapted from public implementations of FermiNet [43] and LapNet [45], both of which are also distributed under the Apache-2.0 License. Modifications were made to these architectures to integrate the $Q^2VMC$ algorithm. Natural gradient updates were based on KFAC-JAX [46], adhering to the same licensing terms. For the LapNet experiments, training was conducted on four Nvidia GeForce 3090 GPUs, utilizing standard single precision calculations and double-precision for matrix multiplications, with training durations ranging from 5 to 90 clock hours depending on the size of the molecule. Similarly, Psiformer experiments were performed in single precision on four Nvidia V100 GPUs, with each run varying from 8 to 140 clock hours.

## 6 Conclusions

In this study, we introduced the Quadratic Quantum Variational Monte Carlo ($Q^2VMC$), which optimizes neural ansatz in quantum variation Monte Carlo by evolving wavefunctions towards the ground state in non-parametric space, then projecting these onto the neural network's parametric manifold using KL divergence minimization. Our experiments demonstrate that $Q^2VMC$ not only strengthens the theoretical foundation but also significantly surpasses traditional QVMC updates in speed and accuracy.

**Limitations and Future Works** While $Q^2VMC$ demonstrates clear advantages, it also faces several unresolved challenges, such as determining the optimal imaginary time step and quantifying the inaccuracies introduced by approximate projection methods. Due to current computational constraints, our experiments were limited to systems with up to 30 electrons. Similarly, these limitations prevented us from conducting a complete set of experiments on other important quantum chemistry applications, such as relative energies [47, 48] and excited states [49–51]. In future work, we aim to extend

$Q^2$VMC to these domains to further assess its performance. Additionally, our method currently achieves only a constant factor speed-up, as observed in the experiments, but the fundamental $\mathcal{O}(N^4)$ scaling with the number of electrons remains a bottleneck, restricting its application to very large systems. We hope to address this scaling issue to enable testing on larger molecules.

**Broader Impacts**  Broader impacts of this work could influence computational chemistry, potentially reducing the reliance on physical experiments and accelerating the discovery of new drugs and environmentally friendly chemical processes, while adhering to stringent ethical standards.

## 7 Acknowledgement

The authors thank the four anonymous reviewers for their invaluable discussions and insightful feedback. Their suggestions regarding limitations and challenges were instrumental in shaping improvements to our paper and inspiring directions for future research. The research is conducted in Statistics & AI group at UT Austin, which receives supports in part from NSF CAREER1846421, SenSE2037267, Office of Navy Research, and NSF AI Institute for Foundations of Machine Learning (IFML).

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

## A   Details of Computing Local Energies and Gradients

**Hamiltonians for Molecules**   The characteristics of a chemical system are encapsulated by its Hamiltonian, which in quantum chemistry typically involves specifying the positions and charges of the atomic nuclei. Under the Born-Oppenheimer approximation, nuclei are treated as classical particles with fixed positions, simplifying the Hamiltonian to:

$$\hat{H} = -\frac{1}{2}\sum_i \nabla_i^2 + \sum_{i>j}\frac{1}{|\mathbf{x}_i - \mathbf{x}_j|} - \sum_{i,I}\frac{Z_I}{|\mathbf{x}_i - \mathbf{X}_I|} + \sum_{I>J}\frac{Z_I Z_J}{|\mathbf{X}_I - \mathbf{X}_J|} \tag{12}$$

Here, $\nabla_i^2$ represents the Laplacian operator for the $i$th electron, and $Z_I$ and $\mathbf{X}_I$ denote the charges and fixed coordinates of the nuclei, respectively. While potential energies are derived straightforwardly, the Laplacian components required for kinetic energy calculations can be efficiently computed using the standard library of JAX [44] or the more advanced Forward Laplacian technique [9].

**Computation of Local Energies**   It is often more stable to parameterize the logarithm of the wavefunction, represented as $f_\theta(\mathbf{x}) = \log\psi_\theta(\mathbf{x})$, rather than the wavefunction itself. The local energies can be expressed entirely in terms of the log wavefunction as follows:

$$E_{L,\theta}(\mathbf{x}) = -\frac{1}{2}\left(\nabla_{\mathbf{x}}^2 \log\psi^2(\mathbf{x}) + \|\nabla_{\mathbf{x}}\log\psi(\mathbf{x})\|^2\right) + V(\mathbf{x}) \tag{13}$$

Here, the gradients and Laplacians are calculated with respect to the positions of all particles, for $i = 1, ..., N$ and across all spatial dimensions $j = 1, 2, 3$. This setup facilitates efficient computation of both the gradient of the log wavefunction with respect to the model parameters $\nabla_\theta \log\psi(\mathbf{x})$ and in combined, the overall updates in both the QVMC and Q$^2$VMC methods, as in equations 4 and **??**.

## B   Derivations of various results presented in the paper

**Theorem B.1.** *Assuming $\langle\psi^{(0)}, \phi_0\rangle \neq 0$ and $\|\psi^{(0)}\|_2 < \infty$, then $\psi^{(n)}$ weakly converges to $\phi_0$, up to a constant factor, as $n \to \infty$.*

*Proof.* Expressing the initial wavefunction in its spectral decomposition form, $\psi^{(0)} = \sum_i \alpha_i\phi_i$, the discretized evolution can be written as:

$$\psi^{(n)} = \sum_i \alpha_i \left(\frac{1 - \tau E_i}{1 - \tau E_0}\right)^n \phi_i = \sum_i \alpha_i^{(n)}\phi_i. \tag{14}$$

Given that $(1 - \tau E_i)/(1 - \tau E_0) < 1$ for $i > 0$ (negative energies), all coefficients $\alpha_i^{(n)}, i > 0$ diminish to zero as $n$ approaches infinity, while $\alpha_0^{(n)}$ remains constant for all $n$. Therefore, the sequence weakly converges to $\alpha_0\phi_0$ by definition. $\square$

**Proposition B.2.** *Let $h(\Delta\theta)$ denote the KL-divergence between the evolved distribution $(1 - \tau E_L(\boldsymbol{x}))^2\psi_\theta^2(\boldsymbol{x})$ and the updated distribution $\psi_{\theta+\Delta\theta}^2$:*

$$h(\Delta\theta) = \mathcal{KL}\left[(1 - \tau E_L(\boldsymbol{x}))^2\psi_\theta^2(\boldsymbol{x})\|\psi_{\theta+\Delta\theta}^2(\boldsymbol{x})\right]. \tag{15}$$

*Given the size of trust region $\epsilon$, our objective of projection is*

$$\Delta\theta_\epsilon^* = \underset{\Delta\theta}{\arg\min}\left\{h(\Delta\theta) \quad s.t. \quad \mathcal{KL}(\psi_{\theta+\Delta\theta}^2\|\psi_\theta^2) \leq \epsilon^2/2\right\}. \tag{16}$$

*As $\epsilon \to 0^+$, the optimal update direction approaches to*

$$\Delta\theta^* = \lim_{\epsilon\to 0^+}\frac{1}{\epsilon}\Delta\theta_\epsilon^* = -\frac{F^{-1}g}{g^\top F^{-1}g} \tag{17}$$

*where $g$ and $F$ are the gradient and Fisher information matrix respectively:*

$$g = \mathbb{E}[\nabla_\theta \log\psi_\theta^2(\boldsymbol{x})] - \left(\mathbb{E}\left[(1 - \tau E_L(\boldsymbol{x}))^2\right]\right)^{-1}\mathbb{E}\left[(1 - \tau E_L(\boldsymbol{x}))^2\nabla_\theta \log\psi_\theta^2(\boldsymbol{x})\right],$$

$$F = \mathbb{E}\left[\left(\nabla_\theta \log \psi_\theta^2 - \mathbb{E}\left[\nabla_\theta \log \psi_\theta^2\right]\right)\left(\nabla_\theta \log \psi_\theta^2 - \mathbb{E}\left[\nabla_\theta \log \psi_\theta^2\right]\right)^\top\right].$$

*represents the Fisher information matrix associated with the distribution induced by the neural ansatz. All expectations here are taken with respect to the distribution $|\psi_\theta(\boldsymbol{x})|^2$.*

*Proof.* Following the results from Section 6 of [42], we establish that for a well-defined objective $h(\theta)$, the optimal update within a trust region is given by:

$$-\frac{F^{-1}g}{g^\top F^{-1}g} = \lim_{\epsilon \to 0} \frac{1}{\epsilon} \arg\min_{\Delta\theta} \left\{ h(\Delta\theta) \quad s.t. \quad \mathcal{KL}(\psi_{\theta+\Delta\theta}^2 \| \psi_\theta^2) \leq \frac{\epsilon^2}{2} \right\} \tag{18}$$

Therefore, the proof left with computing the gradient of the objective and the Fisher information matrix associated with the distribution $|\psi_\theta|^2$.

Let $\theta_0$ represent the fixed part of the parameters, and, with slight abuse of notation, let $\theta$ denotes the variables under optimization. We avoid explicit dependency on $\theta_0$ in the notation of local energies $E_L(\mathbf{x})$ for clarity. The objective function $h(\theta)$ is expressed as:

$$h(\theta) = \mathcal{KL}\left[(1 - \tau E_L(\mathbf{x}))^2 \psi_{\theta_0}^2(\mathbf{x}) \| \psi_{\theta_0+\theta}^2(\mathbf{x})\right]$$
$$= \int \frac{(1 - \tau E_L(\mathbf{x}))^2 \psi_{\theta_0}^2(\mathbf{x})}{\int (1 - \tau E_L(\mathbf{x}))^2 \psi_{\theta_0}^2(\mathbf{x})\mathrm{d}\mathbf{x}} \left(\log \frac{(1 - \tau E_L(\mathbf{x}))^2 \psi_{\theta_0}^2(\mathbf{x})}{\int (1 - \tau E_L(\mathbf{x}))^2 \psi_{\theta_0}^2(\mathbf{x})\mathrm{d}\mathbf{x}} - \log \frac{\psi_{\theta_0+\theta}^2(\mathbf{x})}{\int \psi_{\theta_0+\theta}^2(\mathbf{x})\mathrm{d}\mathbf{x}}\right)\mathrm{d}\mathbf{x}$$

The gradient is then computed as:

$$g = \frac{\partial h}{\partial \theta}\bigg|_{\theta=0}$$
$$= -\frac{\partial}{\partial \theta} \int \frac{(1 - \tau E_L(\mathbf{x}))^2 \psi_{\theta_0}^2(\mathbf{x})}{\int (1 - \tau E_L(\mathbf{x}))^2 \psi_{\theta_0}^2(\mathbf{x})\mathrm{d}\mathbf{x}} \log \frac{\psi_{\theta_0+\theta}^2(\mathbf{x})}{\int \psi_{\theta_0+\theta}^2(\mathbf{x})\mathrm{d}\mathbf{x}}\mathrm{d}\mathbf{x}$$
$$= -\int \frac{(1 - \tau E_L(\mathbf{x}))^2 \psi_{\theta_0}^2(\mathbf{x})}{\int (1 - \tau E_L(\mathbf{x}))^2 \psi_{\theta_0}^2(\mathbf{x})\mathrm{d}\mathbf{x}} \frac{\partial}{\partial \theta} \log \frac{\psi_{\theta_0+\theta}^2(\mathbf{x})}{\int \psi_{\theta_0+\theta}^2(\mathbf{x})\mathrm{d}\mathbf{x}}\mathrm{d}\mathbf{x}$$
$$= \mathbb{E}[\nabla_\theta \log \psi_\theta^2(\mathbf{x})] - \left(\mathbb{E}\left[(1 - \tau E_L(\mathbf{x}))^2\right]\right)^{-1} \mathbb{E}\left[(1 - \tau E_L(\mathbf{x}))^2 \nabla_\theta \log \psi_\theta^2(\mathbf{x})\right]$$

The Fisher information matrix, normalized for the distribution, is:

$$F = \mathbb{E}\left[\left(\nabla_\theta \log \frac{\psi_\theta^2(\mathbf{x})}{\int \psi_\theta^2(\mathbf{x})\mathrm{d}\mathbf{x}}\right)\left(\nabla_\theta \log \frac{\psi_\theta^2(\mathbf{x})}{\int \psi_\theta^2(\mathbf{x})\mathrm{d}\mathbf{x}}\right)^\top\right]$$
$$= \mathbb{E}\left[\left(\nabla_\theta \log \psi_\theta^2 - \mathbb{E}\left[\nabla_\theta \log \psi_\theta^2\right]\right)\left(\nabla_\theta \log \psi_\theta^2 - \mathbb{E}\left[\nabla_\theta \log \psi_\theta^2\right]\right)^\top\right].$$

$\square$

## C   Additional Experiment Results and Hyperparameters

Table 4 details the network architecture hyperparameters for the neural ansatzes Psiformer and LapNet used in our experiments. Table 5 outlines the training hyperparameters employed for pretraining and optimizing these ansatzes. Table 6 presents the results of our efforts to reproduce the baseline energy values, comparing them with the energies reported in the reference papers. These results affirm the consistency of our implementations with those described in the original publications.

Table 5: Neural Network Architecture Hyperparameters

| Parameter | Psiformer | LapNet |
|---|---|---|
| Determinants | 16 | 16 |
| Network layers | 4 | 4 |
| Attention heads | 4 | 4 |
| Attention dims | 64 | 64 |
| MLP hidden dims | 256 | 256 |

Table 6: Optimization Hyperparameters

| Parameter | Value |
|---|---|
| **Training** | |
| Optimizer | KFAC |
| Training iterations | 2e5 |
| Batch size | 4096 |
| Learning rate at iteration $t$ | $lr_0/(1 + t/t_0)$ |
| Initial learning rate $lr_0$ | 0.05 |
| Learning rate decay steps $t_0$ | 1e4 |
| Local energy clipping | 5.0 |
| **Pretraining** | |
| Optimizer | LAMB |
| Pretraining iterations | 2e4 |
| Learning rate | 1e-3 |
| **MCMC** | |
| Decorrelation steps | 30 |
| **KFAC** | |
| Norm constraint | 1e-3 |
| Damping | 1e-3 |
| Momentum | 0 |
| Decay factor of covariance moving average | 0.95 |

Table 7: Energies of reproduced values based on our own experiments comparing the reported baseline values as in [8] and [9]

| System | Psiformer | Psi Reproduced | LapNet | Lap Reproduced |
|---|---|---|---|---|
| $Li_2$ | -14.99486(1) | -14.99488(1) | -14.99485(1) | -14.99486(1) |
| $NH_3$ | -56.56367(2) | -56.56366(2) | -56.56359(2) | -56.56361(2) |
| CO | -113.32416(4) | -113.32429(2) | -113.32417(4) | -113.32416(3) |
| $CH_3NH_2$ | -95.86050(4) | -95.86051(3) | -95.86025(3) | -95.86026(3) |
| $C_2H_6O$ | -155.04656(7) | -155.04667(3) | -155.04563(6) | -155.04561(7) |
| $C_4H_6$ | -155.94619(8) | -155.94618(8) | -155.94528(4) | -155.94535(3) |

Table 8: Energies for molecules tested with Psiformer. The table includes benchmarking values from [8], including both small and large models, where the latter has approximately four times the number of parameters as the former. We present the original results from our paper, obtained using the hyperparameters from [8] (not tuned), as well as results obtained with tuned hyperparameters. Both sets of results are derived from the Psiformer (Small). The results with tuned hyperparameters for the small model match or exceed the accuracies of the benchmarking large model.

| System | Psiformer (Small) | Psiformer (Large) | $Q^2$VMC (original) | $Q^2$VMC (tuned) |
|--------|-------------------|-------------------|---------------------|------------------|
| $Li_2$ | -14.99486(1) | -14.99485(2) | -14.99490(1) | -14.99492(5) |
| $NH_3$ | -56.56367(2) | -56.56381(2) | -56.56374(2) | -56.56386(2) |
| CO | -113.32416(4) | -113.32466(3) | -113.32442(2) | -113.32469(3) |
| $CH_3NH_2$ | -95.86050(4) | -95.86096(3) | -95.86073(2) | -95.86094(3) |
| $C_2H_6O$ | -155.04656(7) | -155.04759(6) | -155.04696(3) | -155.04740(5) |
| $C_4H_6$ | -155.94619(8) | -155.94836(7) | -155.94665(4) | -155.94815(5) |

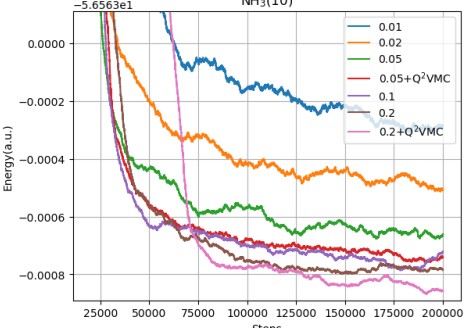

Figure 2: Energy training curves with different learning rates in ablation studies.

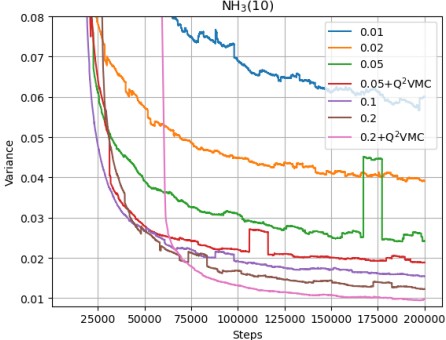

Figure 3: Variance training curves with different learning rates in ablation studies.

