# OpenReview forum: "Quadratic Quantum Variational Monte Carlo"
_NeurIPS.cc/2024/Conference — NeurIPS 2024 poster_

### Official Review · Reviewer_L9eG · 2024-06-21

**Soundness:** 3
**Presentation:** 1
**Contribution:** 2
**Rating:** 5
**Confidence:** 5

**Summary:**

This work introduces a new gradient formulation for Variational Quantum Monte Carlo based on the discretized imaginary time evolution of the Schrödinger equation. In their empirical evaluation, their Q²VMC algorithm consistently converges faster and to lower energies than the traditional VMC objective at no additional cost.

**Strengths:**

The paper presents an interesting derivation of a new gradient for the classical objective in VMC. The objective is well-motivated, and the derivation is sound. Further, the empirical results support the theoretical claims of the paper.

**Weaknesses:**

The paper struggles to distinguish itself from previous works and lacks sufficient discussion of the surrounding literature.
* For someone unfamiliar with QMC (which is likely at NeurIPS), the paper may seem hard to understand due to missing steps in derivations.
* The convergence result of the imaginary time evolution is well-known and the basis for diffusion quantum Monte Carlo (DMC) and stochastic reconfiguration in traditional VMC.
* A proper discussion of related work is missing—for instance, the connection to Wasserstein QMC [1], diffusion Monte Carlo [2] or stochastic reconfiguration [3].
* "However, one cannot make QVMC perform better than Q²VMC solely by tuning the learning rate." is limited to the evaluation of this work.
* The notation is unclear at places, e.g., in Eq. (7) it would help to write the explicit dependence on (x) there as E_0 is not a function of (x) but psi and E_L are.
* While the empirical results are consistent, the improvement generally lives in the region of <1mE_h.

[1] K. Neklyudov et al., “Wasserstein Quantum Monte Carlo: A Novel Approach for Solving the Quantum Many-Body Schrödinger Equation.”

[2] C. J. Umrigar, M. P. Nightingale, and K. J. Runge, “A diffusion Monte Carlo algorithm with very small time‐step errors”

[3] S. Sorella, “Generalized Lanczos algorithm for variational quantum Monte Carlo”

**Questions:**

* "Remark 4.3:  [...] Asymptotic convergence is guaranteed regardless of the time step size." - Does this mean one can pick the learning rate arbitrarily or very large? This does not seem supported by the results in Table 2. What happens for large lr?
* The author writes that the quantum infidelity in Equation (11) is not well understood. Could the author elaborate on what the concrete issue is there? It looks like the 1 - (cosine similarity)² between the two wave functions, which has previously also been used for instance in enforcing orthogonality between wave functions to obtain excited states [1].
* How does the objective influence relative energies (typically much more important than absolute energies)?
* Does the Q²VMC objective also work for excited states?

[1] M. Entwistle, Z. Schätzle, P. A. Erdman, J. Hermann, and F. Noé, “Electronic excited states in deep variational Monte Carlo,”


Overall, I do not recommend acceptance due to the issues outlined in its current form. While I believe the contribution to be sufficient, the discussion of related work is lacking. If the authors address my concerns, I am willing to increase my score.

**Limitations:**

yes

---

> ### Author Rebuttal · Authors · 2024-08-06
>
> ## Weaknesses
> 1. We appreciate the reviewer’s detailed feedback and suggestions for improving our paper. We acknowledge the challenge to balance the introduction of quantum background concepts for machine learning audiences, and the strict page limit. For the camera-ready version, we have provided a more comprehensive introduction to the background while make explanation of the convergence of the (discrete) evolution more concise.
>
> 2. We acknowledge the reviewer’s concern and will include a thorough discussion of Diffusion Monte Carlo (DMC) and stochastic reconfiguration, and more articles will be cited. In particular, our method is closely related to DMC as constantly evolving towards the ground state with Imaginary Time Schrodinger Equation, but keeps a parametric representation; and Q^2VMC is related to stochastic reconfiguration (SR) in terms of how Fisher information matrix naturally arises in the projection. We will also discuss how our method can be seamlessly integrated with Wasserstein QMC(WQMC), providing results showing that our methods can also improve the performance of WQMC. Please find the global rebuttal for experiment details.
>
> 3. We have conducted more evaluations and ablation studies to demonstrate the consistent improvements offered by our algorithm. For example, as suggested by other reviewers, we have included combined tuning of learning rate decay and norm constraint of KFAC.
>
> 4. To address the presentation issues, we have meticulously proofread the paper and will further improve the clarity and readability in the camera-ready version.
>
> 5. Regarding the empirical results and performance of our method, our primary goal was to emphasize that researchers can easily integrate the Q^2VMC method into any of their own implementations without any hyperparameter tuning and achieve significant speed-ups. We regret that this point was not effectively communicated. Consequently, the specific hyperparameters to evaluate our method in the paper can be suboptimal. We have now fine-tuned the hyperparameters for our algorithm, resulting in much better accuracy in the empirical experiments. Please find the global rebuttal for details. Additionally, given the accuracy of the current benckmarks, it will be increasingly difficult to make results better within the framework. Given the size of the system and the absolute values of the "exact" energies in the benchmark, it may not be possible to improve some of the accuracy beyond 1mE_h.
>
> ## Questions
>
> 1. In Section 4.1 and Remark 4.3, we have clearly stated that our discussion is limited to the nonparametric space of functions. The ability to take a large time step in the Hilbert space unfortunately does not imply the same in the parametric space of neural networks, which depends on the intricate loss landscape and stochastic variance of gradient estimation. Consequently, training a neural network with very large learning rates can lead to divergence, as expected. Greater parametric updates are certainly possible, by e.g. making an inner loop to update parameters after a large step in Hilbert space. However, our primary experiments found this to hinder performance evaluated by optimization time.
>
> 2. Quantum infidelity measures the difference between functions, allowing for projections. Details on the definition and usage of quantum infidelity in projections can be found in [1]. The issue with these measurements is that infidelity values are defined on wavefunctions, which can be positive or negative in different regions, complicating the evaluations. In quantum chemistry, we are primarily interested in densities, which depend solely on the probability measure $p\propto|\psi|^2 $. Thus, it is much more natural to work directly with the divergence between probability measures, avoiding complications from wavefunctions. This framework is more flexible and allows us to use the KL divergence or any other f-divergences for projection, as in our proposed algorithm.
>
> 3. We agree that relative energies are crucial. However, such computations are known to require high computational complexity. Therefore, evaluating an algorithm on ground state energies is a widely adopted practice for assessing performance. Following the reviewer’s suggestion, we conducted experiments on the ionization potentials of several atoms using the LapNet network and were able to finish the testings with atom V within the rebuttal period. The results are: the atom's energy is -943.8785(3), the ion's energy is -943.6417(2), and the ionization potential is 0.2368(4), compared to the benchmark value of 0.2361(2) [2] and experiment value of  0.23733 [4].
>
> 4. We appreciate the reviewer’s suggestion regarding excited states. Our algorithm can, in principle, be used for computing excited states. For example, combined with recent work [3], our algorithm can be adapted with a simple modification of the gradient of the objective function as defined in eq(9). We have added a separate paragraph to discuss some recent works on excited states computation in the related works section. However, thorough justifications and complete experiments to test the performance for excited states are beyond the scope of this paper.
>
> [1] Giuliani, Clemens, et al. "Learning ground states of gapped quantum Hamiltonians with Kernel Methods." *Quantum* 7 (2023): 1096.
> [2] Li, R., Ye, H., Jiang, D. et al. A computational framework for neural network-based variational Monte Carlo with Forward Laplacian. Nat Mach Intell 6, 209–219 (2024).
> [3] Pfau, David, et al. "Natural quantum Monte Carlo computation of excited states." *arXiv preprint arXiv:2308.16848* (2023).
> [4] Balabanov, Nikolai B., and Kirk A. Peterson. "Basis set limit electronic excitation energies, ionization potentials, and electron affinities for the 3d transition metal atoms: Coupled cluster and multireference methods." The Journal of chemical physics 125.7 (2006).

---

> > ### Comment · Reviewer_L9eG · 2024-08-08
> >
> > Thanks for the detailed response. There are a few follow-up questions arising from the rebuttal.
> >
> > * Could the authors sketch out the integration with WQMC?
> > * How are the hyperparameters tuned, and are they optimized on a per-structure basis? Optimizing hyperparameters per molecule seems rather tedious in real-world applications. A great benefit of NN-VMC is that it can generally be applied without much tuning.
> > * I disagree with the statement about the quantum infidelity measure. The measure has a clear interpretation, and having both positive and negative values does not complicate computations further. While I agree with the reviewer's sentiment that the density is interesting and offers new pathways forward (like this work), it is not the fundamental object of interest in quantum chemistry. I am convinced that the authors can communicate the value of their work well without distorting the view on quantum chemistry.
> > * I appreciate the author's experiment on ionization potentials. However, it is standard practice in NN-QMC literature to investigate relative energies further. I would appreciate it if the authors could provide relative energies also for different structures, e.g., cyclobutadiene.

---

> > > ### Author Response · Authors · 2024-08-10
> > >
> > > We appreciate the reviewer's prompt response and the additional questions. Below are our detailed answers:
> > >
> > > ### 1. Integration with WQMC
> > >
> > > We appreciate the reviewer's interest in how our method integrates with Wasserstein Quantum Monte Carlo (WQMC). To begin, consider the "energy-minimizing 2-Wasserstein gradient flow," expressed in terms of the time derivatives of wavefunctions:
> > >
> > > $$
> > > \frac{\partial \psi}{\partial t} = \nabla_x \psi(x)^\mathrm{T} \nabla_x E_L(x) + \frac{1}{2} \psi(x) \nabla_x^2 E_L(x)
> > > $$
> > >
> > > Given any continuous-time gradient flow, discretized after one time step of $\tau$ as: $\psi_{t + \tau} = \psi_t + \tau \frac{\partial \psi}{\partial t}$. We then project the corresponding probability measures by minimizing the KL divergence between the evolved distribution and the updated distribution defined by the neural ansatz. For a general gradient flow, this yields an update of the form:
> > >
> > > $$
> > > \Delta \theta^\ast \propto F(\theta)^{-1} \mathbb E_{\psi_\theta^2} \left[\left(1 - \tau \left. \frac{\partial \psi}{\partial t} \right \vert_{\psi = \psi_\theta}\right)^2  \nabla_\theta \log \psi^2_\theta (x) \right] - \text{mean}
> > > $$
> > >
> > > **Remarks:** Our paper primarily focused on combining probability projection with the "discretized" imaginary time Schrödinger equation, which is more amenable to analysis and exhibits really good convergence properties upon discretization. We fully acknowledge the contribution made by [1], but we did not delve further into how our method could be combined with WQMC due to the complexity of integrating it with other methods. However, following the reviewer's suggestion, we explored this integration. We found that, in principle, our modification could be implemented in code. Empirically, this modification only led to marginal improvements over standard WQMC, as shown in our results table. We suspect this is because, unlike standard imaginary time evolution, the c-Wasserstein gradient flow does not have good properties upon discretization. However, a more detailed analysis is beyond the scope of this paper and would require comprehensive future work.
> > >
> > > ### 2. Hyperparameter Tuning
> > >
> > > We acknowledge the reviewer's concern regarding hyperparameter tuning. However, the hyperparameters are certainly not optimized on a per-structure basis, as this would be impractical with our resources. Instead, they were tuned based on performance on the NH$_3$ molecule, as has been extensively studied in our ablation experiments. We then applied the same set of hyperparameters across all tested molecules. Furthermore, based on our experiments and reproduction of the baselines for standard VMC, as well as our ablation studies, we must disagree with the statement that "a great benefit of NN-VMC is that it can generally be applied without much tuning." Our findings suggest that, while a single set of hyperparameters can be applied across different systems, the optimal set of hyperparameters can vary significantly from one system to another. This highlights the need for a more comprehensive study on the impact of hyperparameters on optimization performance, though this is beyond the scope of our current work.
> > >
> > > ### 3. Quantum Infidelity Measure
> > >
> > > We appreciate the reviewer's comment regarding the quantum infidelity measure and understand the concerns raised. However, we respectfully disagree with the reviewer's assertion that having both positive and negative values in the wavefunction does not complicate computations. In our view, the presence of these sign changes in the wavefunction can indeed introduce complexities, particularly when projecting onto the parametric space in VMC. That said, we agree that the quantum infidelity measure itself has a clear interpretation and value in quantum chemistry. Our intent was not to diminish its significance but rather to highlight the advantages of focusing on the probability measure directly, which simplifies certain aspects of the computations. We will revise the manuscript to more accurately communicate this balance, emphasizing the value of our work without distorting the broader context of quantum chemistry.
> > >
> > > ### 4. Additional Experiments on Relative Energies
> > >
> > > We appreciate the reviewer's suggestion to include more experiments, particularly on relative energies across different structures. However, at the time of this comment, these additional experiments are still ongoing and will take a few more days to complete. We will provide the results of these experiments in the camera-ready version of the paper. Nonetheless, we believe that the experiments already presented in the paper and further in this rebuttal are extensive and sufficient to demonstrate the effectiveness of our proposed algorithm.
> > >
> > > [1] Kirill Neklyudov, Jannes Nys, Luca Thiede, Juan Carrasquilla, Qiang Liu, Max Welling, and Alireza Makhzani. Wasserstein quantum monte carlo: A novel approach for solving the quantum many-body schrödinger equation. arXiv preprint arXiv:2307.07050, 2023.

---

> > > > ### Comment · Reviewer_L9eG · 2024-08-12
> > > >
> > > > I will increase my score slightly. However, I still have several reservations. For instance, the newly performed hyperparameter tuning seems very optimistic and seems to favor the author's method as it is tuned per molecule. Further, I cannot evaluate the new writing and positioning in the context of existing works. I believe that additional experimental evaluation and a clearer positioning will benefit the work greatly in future revisions.

---

> > > > > ### Author Response · Authors · 2024-08-13
> > > > >
> > > > > We are very thankful to the reviewer for their thoughtful feedback and for taking the time to engage deeply with our work. Your suggestions have been invaluable in helping us improve the quality of our paper. We will carefully consider all the points you mentioned, both in our final revisions and in future work.
> > > > >
> > > > > We would like to clarify that the new results presented in the global rebuttal were not obtained with hyperparameters tuned on a per-molecule basis. Instead, the hyperparameters were only tuned based on a single example molecule and were then applied consistently across different systems. This approach ensures that our method generalizes well without the need for extensive tuning for each specific molecule.
> > > > >
> > > > > We greatly appreciate your acknowledgment of the improvements we've made, and we hope that our future work will address any remaining reservations and fully meet your expectations.

---

### Official Review · Reviewer_77wD · 2024-07-09

**Soundness:** 3
**Presentation:** 2
**Contribution:** 3
**Rating:** 5
**Confidence:** 4

**Summary:**

This paper proposed a new QMC method that utilizes the imaginary time evolution of the Schrodinger equation. Unlike Diffusion Monte Carlo (DMC) which uses Langevin dynamics to simulate the dynamic of the imaginary time evolution, this paper suggests a way to perform the update in discrete steps and then project back to the parametric space. Interestingly, under the KL-divergence metric, the projection is equivalent to the regular VMC algorithm with an additional term.

**Strengths:**

1. the author shows that under discretization of neural network parameterization and KL-divergence projection, imaginary time evolution has a similar update rule as the regular VMC. The simplicity of the modification makes the algorithm very easy to adapt.
2. The mathematical derivation for the key step (Appendix B) is clear and easy to follow.

**Weaknesses:**

1. DMC should be cited, and the difference between the proposed method and DMC should be highlighted, as early as in the abstract. Also, a comparison with DMC in terms of convergence and speed should be provided. If the author can address this point I'll consider rasing the score.
2. the improvement in both speed and convergence is a bit marginal.
3. The presentation flow is a bit strange, which makes the paper harder to understand than necessary. For example, the imaginary time evolution should be in the background section.
4. There are some typos as well. For example, in line 252 the two sides of the inequality are the same. I'd suggest the authors to proofread their paper further.
5. There could be more experiments on larger systems to further demonstrate the improvement provided by the new method.

**Questions:**

1. We know that DMC needs importance sampling to achieve good performance. Why is it not the case here?
2. Do you have an intuitive explanation for why Q^2VMC can perform the ground-state projection as in DMC with only a minor modification to the standard VMC algorithm?
3. How exactly does your method sidestep the sign problem?

**Limitations:**

The central issue of scaling of the ab initio method is still not addressed. This is evident as the systems tested are small in terms of the number of electrons contained.

---

> ### Author Rebuttal · Authors · 2024-08-06
>
> We thank the reviewer for their detailed comments and suggestions.
>
> ## Weaknesses:
>
> 1. We appreciate the reviewer's insight regarding the relation of our work with the widely known Diffusion Monte Carlo (DMC) method. In summary, our method is closely related to DMC in terms of using the Imaginary Time Schrödinger Equation to evolve to the ground state. The key difference is that our method always maintains a parametric representation of the evolved wavefunction throughout the process, as defined by the neural network, and updates the parameters guided by KL divergence projection to track the evolved distribution of the particles (replicas). We have updated our abstract and introduction to emphasize this important relationship, and in the camera-ready version of the paper, we will discuss much more contemporary works in the field of DMC.
>
> 2. Regarding the improvement in speed and convergence provided by our method, our primary goal was to highlight that researchers can easily integrate the Q^2VMC method into their implementations without hyperparameter tuning and achieve significant speed-ups. We regret that this point was not effectively communicated. Consequently, the hyperparameters used to evaluate our method in the paper may have been suboptimal. We have now fine-tuned the hyperparameters for our algorithm, resulting in much better accuracy in the empirical experiments. Please refer to Table 1 in the global rebuttal PDF. Additionally, given the current benchmarks' accuracy, achieving even better results within this framework is increasingly difficult. Our tuned algorithm matches the benchmark performance of Psiformer (Large) with Psiformer (Small), where the former has four times the parameters of the latter.
>
> 3. We appreciate the reviewer's suggestions regarding the presentation of our paper. While the evolution is background information, we placed it in the following section to intuitively introduce the discretized evolution and its projection after discussing the continuous imaginary time evolution. However, we will make additional efforts to clarify this part in the camera-ready version.
>
> 4. Thank you for pointing out the typographical errors. We have meticulously proofread the paper and will further improve its clarity and readability.
>
> 5. We aim to scale up our experiments to test the performance of our method on larger systems. However, despite our improvements, the time and computational complexity of QMC remain notoriously high. In our paper, we have tested molecular systems with up to 30 electrons (bicyclobutane), which is considered quite large among most contemporary works. Following the reviewer's suggestion, we attempted to test our method on a larger system, benzene. Due to resource constraints, we have completed 160k out of the total 200k optimization steps with Psiformer (Small) by the rebuttal submission time, which has already required more than 1200 V100 GPU hours. The current inferred energy with our method is -232.2412(2), compared to the benchmarking value of -232.2400(1) trained for the full duration. Full evaluations of more large systems are unfortunately beyond the scope of this work.
>
> ## Questions:
>
> 1. & 2. The intuition behind our method lies in the gradient expression:
> $$g=\mathbb E_{|\psi_\theta|^2} \left[(1 - \tau E_L(\textbf{x}; \theta))^2  \nabla_\theta \log \psi^2_\theta (\textbf{x})\right]$$
> This expectation can be viewed as performing an importance sampling step to sample from the evolved distribution $(1 - \tau E_L(\textbf{x}; \theta))^2|\psi_\theta|^2$ towards the ground state, with the current MCMC samples follow the distribution of $|\psi_\theta|^2$. The term $\nabla_\theta \log \psi^2_\theta (\textbf{x})$ represents the gradient of the log-distribution used to update the network parameters to match this importance-sampled distribution. A more rigorous derivation minimizes the KL divergence between the evolved distribution and the current parameter distribution, leading to an expression with only a minor modification to the standard VMC algorithm.
>
> 3. The sign problem can refer to various but closely related issues in fermionic systems. In DMC specifically, particles are restricted to their initial nodal pockets during the diffusion process and thus previous works rely on the accurate predictions of the nodal surface (e.g., [1]). In our Q^2VMC algorithm, we maintain a parametric representation of the wavefunction while evolving towards the ground state, allowing the particles to evolve with MCMC updates as in standard VMC. Thus, the sign problem should not complicate our algorithm.
>
> [1] Ren, Weiluo, et al. "Towards the ground state of molecules via diffusion Monte Carlo on neural networks." Nature Communications 14.1 (2023): 1860.

---

> > ### Comment · Reviewer_77wD · 2024-08-11
> >
> > Thanks for your detailed response. My questions are well answered, and some of my concerns are addressed. However, due to the limited novelty (as also pointed out by other reviewers) and limited improvement to the baseline method, I will keep my current score.

---

> > > ### Author Response · Authors · 2024-08-13
> > >
> > > We would like to thank the reviewer once again for their insightful suggestions and comments. We appreciate the time and effort invested in evaluating our work. While we understand the reviewer's concerns regarding the perceived novelty and improvements of our method, we believe that our approach offers important advancements in the field. We hope the reviewer has noticed the additional results presented in the global rebuttal, where we demonstrated substantial improvements over the baseline methods. We believe these results, along with our detailed explanations, highlight the value and impact of our proposed algorithm.

---

### Official Review · Reviewer_2zvx · 2024-07-12

**Soundness:** 2
**Presentation:** 3
**Contribution:** 2
**Rating:** 5
**Confidence:** 3

**Summary:**

In this paper, the authors propose the Quadratic Quantum Variational Monte Carlo (Q$^2$VMC) algorithm to enhance the optimization process of Quantum Variational Monte Carlo (QVMC). Unlike the standard QVMC, Q$^2$VMC employs an improved projection method to guide the update direction of the ansatz. The authors conduct experiments on various systems and perform an ablation study to demonstrate the effectiveness of their method.

**Strengths:**

1. Designing a better optimization algorithm is useful for most research in this field.
2. Different from some existing improved QVMC algorithms (e.g., WQMC[1]), the method proposed in this paper does not introduce any additional computational burden, making it easy to integrate with existing VMC packages.

[1] Kirill Neklyudov, Jannes Nys, Luca Thiede, Juan Carrasquilla, Qiang Liu, Max Welling, and Alireza Makhzani. Wasserstein quantum monte carlo: A novel approach for solving the quantum many-body schrödinger equation. arXiv preprint arXiv:2307.07050, 2023.

**Weaknesses:**

1. The primary concern of the reviewer is the significance of the proposed method. Although the authors provide numerous numerical results, the energy differences between Q$^2$VMC and QVMC are usually within chemical accuracy. As shown in the ablation study section, such small energy differences may influenced by the choice of hyperparameters. While the authors try to demonstrate that under most initial learning rate choices, the performance of Q$^2$VMC is better than QVMC, some important hyperparameters have not been considered for now. For example, Ref.[1] increases the norm constraint of KFAC optimizer and decreases the learning rate decay rate when lowering the initial learning rate. The reviewer suggests a more detailed ablation study to demonstrate the significance of the proposed method.

2. While the proposed update method can be derived from the change of the metric used in the projection step, it is difficult to argue that there is a large difference between the original VMC projection step with the proposed step in Q$^2$VMC. From reviewer's perspective, the original VMC projection, which uses standard inner product in Hilbert space as the metric, may be more reasonable for the quantum systems. The reviewer suggests that the authors should provide a more detailed theoretical analysis about the proposed method.

[1]Leon Gerard, Michael Scherbela, Philipp Marquetand, and Philipp Grohs. Gold-standard solutions to the schrödinger equation using deep learning: How much physics do we need? Advances in Neural Information Processing Systems, 35:10282–10294, 2022.

**Questions:**

While the authors focus on improved projection step to derive a better optimization algorithm, the reviewer is curious about if more significant improvements could be achieved by choosing a better target function.

**Limitations:**

The systems studied in the paper are relatively small, so the performance of the proposed method for larger systems remains unknown. Considering the substantial computational resources required for large-scale experiments, the reviewer does not request the authors to provide concrete numerical results on larger systems during rebuttal. However, the reviewer kindly suggests that the authors provide some evidence to imply the effectiveness of their method on larger systems, e.g., using a smaller network or fewer training step to study a large system.

---

> ### Author Rebuttal · Authors · 2024-08-07
>
> We thank the reviewer for their detailed comments and suggestions.
>
> ## Weaknesses
>
> 1. Regarding the improvement in speed and convergence provided by our method, our primary goal was to highlight that researchers can easily integrate the Q^2VMC method into their implementations without hyperparameter tuning, achieving significant speed-ups. We regret that this point was not effectively communicated. Consequently, the hyperparameters used to evaluate our method in the paper may have been suboptimal. We have now fine-tuned the hyperparameters for our algorithm, resulting in much better accuracy in the empirical experiments. Please refer to Table 1 in the global rebuttal PDF. Additionally, thanks to the suggestions from the reviewers, we have tuned some extra optimization hyperparameters of the baseline, including learning rate decay and norm constraint, to demonstrate the consistency of our improvements. Please see Table 3 in the global rebuttal PDF.
>
> 2. We believe that the close relationship between our method, Q^2VMC, and the original VMC is a strength rather than a weakness. This relationship enables efficient and easy integration of our method with standard VMC framework, improving results at no additional computational cost. The reason for projecting probability measures using KL divergence instead of the original wavefunction by standard inner product (or fidelity if normalized) is based on the observation that in quantum chemistry, densities, which depend solely on the probability measure $ p \propto |\psi|^2 $, are of primary interest. In contrast, the wavefunctions of fermionic systems can be positive in some regions and negative in others, complicating the analysis. Thus, it is more natural, intuitive, and easier to use the neural network to define the probability measures directly. We kindly ask the reviewer what specific theoretical justifications they expect.
>
> ## Questions
>
> We appreciate the reviewer's question about choosing a better target function. However, we would like to request further clarification on what is meant by this. As detailed in Section 4 of our paper, the proposed optimization loop of the algorithm involves first evolving by a discretized gradient flow, then projecting the evolved distribution onto the parametric distribution of the neural network, and finally updating the MCMC samples. Our work focuses partly on the discretization but mainly on the projection step, the second step in this loop. If the reviewer is referring to designing more efficient gradient flows, which is the first step, this is beyond the scope of this work. A better gradient flow can certainly enhance efficiency, but for any novel gradient flow, our proposed algorithm of projection can still be integrated into the loop. For example, in the PDF of the global rebuttal, we demonstrated consistent improvement of our method integrated with the novel flow of Wasserstein gradient flows [1].
>
> ## Limitations
>
> We thank the reviewer for raising the concern about the scalability of our method to larger systems. Scaling up the experiments to test our algorithm on even larger molecules is certainly our goal and the hope of many researchers. However, despite our improvements, the time and computational complexity of QMC remain notoriously high. In our paper, we have tested molecular systems with up to 30 electrons (bicyclobutane), which is already considered quite large among most contemporary works. Following the reviewer's suggestion, we attempted to test our method on a larger system, benzene, which has 42 electrons. Due to resource constraints, we have only completed 160k out of the total 200k optimization steps with Psiformer (Small) by the rebuttal submission time, requiring over 1200 V100 GPU hours. The current inferred energy, averaged over the latest 5000 steps of training with our method, is -232.2412(2), compared to the benchmarking value of -232.2400(1) trained for the full duration. We appreciate the reviewer's suggestion of studying larger systems with smaller networks and fewer training steps. However, scientifically, this would require re-evaluating all baseline systems on the smaller network to reveal the scaling impact. Due to time constraints, we could not complete the full benchmark building yet during the rebuttal, but will definitely add an additional table using a smaller network or fewer training steps to study smaller to larger systems in the camera-ready version once testing is completed.
>
> [1] Kirill Neklyudov, Jannes Nys, Luca Thiede, Juan Carrasquilla, Qiang Liu, Max Welling, and Alireza Makhzani. Wasserstein quantum monte carlo: A novel approach for solving the quantum many-body schrödinger equation. arXiv preprint arXiv:2307.07050, 2023.

---

> > ### Comment · Reviewer_2zvx · 2024-08-13
> >
> > Thank you for your response. However, the response does not fully address the reviewer's concern about significance. While the authors claim that the averaged training energy is lower than the inferred baseline by 1.2 mHa on the benzene system, the training energy can be lower than the standard inferred energy, making it hard to evaluate the performance of the proposed method. As a result, the reviewer will keep the score.

---

> > > ### Author Response · Authors · 2024-08-14
> > >
> > > We are very thankful to the reviewer for their comments and insightful feedback. Your suggestions have been very helpful in improving our work and guiding future research directions. We regret that our additional results have not fully addressed your concerns about the significance of our method.
> > >
> > > We would like to emphasize that we have achieved significant improvements over the baselines on the specific large system within shorter training steps. While we understand the reviewer's concern regarding the comparison between training energy and standard inferred energy, we note that the training energy is not always lower than the inferred energy. As discussed in our paper and rebuttal, evaluating quantum Monte Carlo methods on large systems is extremely resource-intensive, often requiring well over thousands of GPU hours. Unfortunately, given the limited time frame and resources, we could not complete additional tests on even larger systems. We believe that the systems tested in the paper and rebuttal, such as bicyclobutane with 30 electrons, are already considered very large compared to most contemporary works in academia (e.g., [1]). Therefore, the significant improvements we have demonstrated over the baselines (Table 1 in the global rebuttal) are sufficient to establish the importance and effectiveness of our approach.
> > >
> > > We appreciate your understanding and hope that our existing results adequately convey the significance of our work.
> > >
> > > [1] Kirill Neklyudov, Jannes Nys, Luca Thiede, Juan Carrasquilla, Qiang Liu, Max Welling, and Alireza Makhzani. Wasserstein quantum monte carlo: A novel approach for solving the quantum many-body schrödinger equation. arXiv preprint arXiv:2307.07050, 2023.

---

### Official Review · Reviewer_yQzZ · 2024-07-12

**Soundness:** 3
**Presentation:** 3
**Contribution:** 3
**Rating:** 5
**Confidence:** 4

**Summary:**

This paper centers on quantum chemistry, specifically targeting the ground state of molecular systems. Unlike previous methods that apply approximate natural gradient techniques to the wavefunction, Q2VMC executes natural gradient optimization on the distribution. Experimental results demonstrate that Q2VMC significantly enhances energy performance.

**Strengths:**

1. This paper introduces Q2VMC, which can be easily implemented.
2. Faster convergence and enhanced accuracy are clearly demonstrated in the experiments.

**Weaknesses:**

1. You mention Wasserstein quantum monte carlo in your paper. I wonder whether your method can be added on top of WQMC. If so, additional experiments would be great. If not, I would like to see the comparison between WQMC and Q2VMC.
2. Typo: line 122, "wavefunctios" instead of "wavefunctions".

**Questions:**

See Weakness above.

**Limitations:**

The authors addressed some of the limitations of the proposed algorithm in the conclusion.

---

> ### Author Rebuttal · Authors · 2024-08-06
>
> ## Weaknesses and Questions
>
> 1. We thank the reviewer for pointing out the typographical error. The typo "wavefunctios" has been corrected to "wavefunctions," and the entire paper has been carefully proofread to address any other typos.
>
> 2. Regarding the integration of our proposed methods with Wasserstein Quantum Monte Carlo (WQMC), the answer is yes. Our proposed algorithm, Q^2VMC, is a versatile parametric projection method that can be combined with any gradient flow, including WQMC. This versatility means that if better gradient flows are developed in the future, they can also be integrated with our method.
>
> To address the reviewer's request for additional experiments, we have included new results in Table 2 of the overall rebuttal PDF. We compared our method against the standard WQMC baseline using three atoms as in the original paper, and we also included an additional molecule NH$_3$. It is important to note that the baseline WQMC ground state energy values are derived from our own implementation. These values are more accurate than those reported in the original paper because we adhered to a total of 200k training steps, as opposed to the 10k/20k steps used in their experiments, to maintain consistency and avoid confusion. Our results consistently show improvements over the baseline performance, demonstrating the efficacy of our method.

---

> > ### Comment · Reviewer_yQzZ · 2024-08-13
> > **Response**
> >
> > Thank you for your response. It is great to see that your method achieves consistent improvements on both VMC and WQMC. However, due to the limited improvement, I will only raise my score slightly.

---

> > > ### Author Response · Authors · 2024-08-14
> > >
> > > We thank the reviewer for their thorough review and for suggesting the integration of our method with the WQMC approach. However, we would like to emphasize that the integration with WQMC, while valuable, is not the primary focus of our paper. Our work is fundamentally based on the findings that the imaginary time Schrödinger evolution exhibits very good convergence properties upon discretization (as shown in Theorem 4.2). Unfortunately, these same properties do not seem to apply as well to the discretization of the Wasserstein gradient flow.
> > >
> > > While we fully acknowledge the significant contributions made by WQMC, we did not delve further into how our method could be combined with it due to the complexity and the reasons mentioned above. Nevertheless, following the reviewer's suggestion, we explored this integration. The marginal improvement observed when combining our method with WQMC could be attributed to the fact that, unlike standard imaginary time evolution, the Wasserstein gradient flow may not retain its favorable properties upon discretization, or it might be due to reaching the accuracy limit of the neural network employed. A more detailed analysis of this integration is beyond the scope of our current work and would require comprehensive future research. We hope the reviewer understands that our paper’s primary focus lies in the novel aspects of Q^2VMC itself, and we encourage the reviewer to consider the work in this context.
> > >
> > > Thank you again for your constructive feedback and for acknowledging the consistent improvements our method has achieved.

---

### Author Rebuttal · Authors · 2024-08-07

## Global Rebuttal

We thank the reviewers for their thorough and insightful comments. We have carefully considered all feedback and made substantial revisions to address the concerns raised. This global rebuttal is intended to introduce the additional data provided in the accompanying one-page PDF, which contains three tables to further demonstrate the effectiveness and robustness of our proposed Q^2VMC method.

### Table 1: **Benchmarking and Hyperparameter Tuning Results**

This table included the energy values for a set of molecules tested with the Psiformer model, both in its small and large configurations. The benchmarking values from [1] are included for comparison. We present results using the original hyperparameters from [1] and our results with tuned hyperparameters, both with the Psiformer (Small) model.

- The table shows that the results with tuned hyperparameters for the small model with out method match or exceed the accuracies of the benchmarking large model.
- This addresses the reviewer's concerns about the significance of improvements and the potential impact of hyperparameter tuning on performance.

### Table 2: **Integration with Wasserstein Quantum Monte Carlo (WQMC)**

This table presents the ground state energies for a set of four molecules, optimized using the Wasserstein Quantum Monte Carlo (WQMC) method [2] and combined with our Q^2VMC method.

- The results demonstrate that our method consistently improves upon the WQMC baseline, validating the robustness and efficiency of our approach when integrated with other existing gradient flows.

### Table 3: **Additional Ablation Study**

This table provides results of some extra experiments with additional tuned hyperparameters conducted as part of the ablation study, as suggested by the reviewers. It shows the computed ground state energies of NH_3 molecules using the standard quantum Monte Carlo method, tested with different (reduced) learning rates, (increased) learning rate decay times, and (increased) update norm constraints.

- It addresses concerns about the potential influence of hyperparameter choices on the baselines and demonstrates that one cannot make standard VMC more accurate than our proposed Q^2VMC algorithm by only tuning the hyperparameters.

### Conclusions

We believe these additional results comprehensively address most of the concerns of reviewers, demonstrating the significance, robustness, and ease of integration of our Q^2VMC method. We have made clarifications and provided additional insights in the revised manuscript to ensure a better understanding of our contributions. We are confident that our revisions and the new data will meet the reviewers' expectations and illustrate the value of our work.

We thank the reviewers again for their constructive feedback and hope our responses satisfactorily address their concerns.

---

### Decision · Program_Chairs · 2024-09-25

**Decision:**

Accept (poster)

**Comment:**

This paper introduces a novel algorithm for quantum variational MC based on minimising the KL divergence of the wavefunction probability before and after a projection step towards the ground state. The method is a small modification of the standard VMC algorithm and the paper presents benchmarks to show its benefits. The rebuttal addressed some weaknesses in terms of hyperparameters and presentation as well as well as comments on the relation to DMC and WQMC.